# Response of cbbL Carbon-Sequestering Microorganisms to Simulated Warming in the River Source Wetland of the Wayan Mountains

**DOI:** 10.3390/biology14060708

**Published:** 2025-06-16

**Authors:** Shijia Zhou, Kelong Chen, Ni Zhang, Siyu Wang, Zhiyun Zhou, Jianqing Sun

**Affiliations:** 1Qinghai Province Key Laboratory of Physical Geography and Environmental Process, College of Geographical Science, Qinghai Normal University, Xining 810008, China; ruye1125@163.com (S.Z.); 15229584811@163.com (S.W.); 13897423633@163.com (Z.Z.); 2Key Laboratory of the Qinghai–Tibet Plateau Land Surface Processes and Ecological Conservation, Ministry of Education, Qinghai Normal University, Xining 810008, China; 3National Positioning Observation and Research Station of Qinghai Lake Wetland Ecosystem in Qinghai, National Forestry and Grassland Administration, Haibei 812300, China; 4Qinghai Lake National Nature Reserve Administration, Xining 810008, China; sunjq@163.com

**Keywords:** Qinghai–Tibet Plateau, river source wetlands, carbon sequestration microorganisms, simulated warming

## Abstract

This study investigated the structural shifts in carbon-sequestering microbial communities and associated soil physicochemical properties in the river source wetland of the Wayan Mountains under simulated warming scenarios. Our findings demonstrate that experimental warming significantly enhanced the α-diversity of soil carbon-fixing microorganisms (*p* < 0.05), while community evenness indices remained statistically unaltered. Notably, we observed substantial enrichment of Cyanobacteria and Actinobacteria phyla, whereas Proteobacteria maintained its ecological dominance under elevated temperatures. The proliferation of keystone genera *Thioflexothrix* and *Ferrithrix* exhibited strong correlations with increased total carbon (TC) and total nitrogen (TN) concentrations (R^2^ > 0.65, *p* < 0.01), underscoring carbon and nitrogen availability as pivotal drivers of microbial community succession in these wetland ecosystems.

## 1. Introduction

Anthropogenic activities, particularly fossil fuel combustion and land-use modifications, have driven unprecedented increases in atmospheric greenhouse gas concentrations, accelerating global warming trends [1]. Projections indicate that by 2100, the global mean annual temperature could exceed 4 °C, profoundly destabilizing soil organic matter (SOM) stocks and disrupting the global carbon cycle [2,3]. Wetlands, recognized as critical global carbon sinks, play a disproportionally large role in carbon sequestration, storing approximately one-third of the world’s soil organic carbon reservoirs [4]. Their anoxic conditions inherently suppress microbial decomposition, facilitating long-term carbon accumulation [4]. However, climate-induced perturbations threaten this balance: elevated temperatures may amplify microbial mineralization of organic substrates, potentially triggering massive CO_2_ and CH_4_ emissions that exacerbate planetary warming [5]. These processes underscore the urgency of understanding soil respiratory responses, as terrestrial soil respiration currently transfers carbon to the atmosphere at rates ~10-fold, exceeding anthropogenic emissions [6]. Given their disproportionate carbon storage capacity and vulnerability, wetland ecosystems constitute vital components for maintaining terrestrial carbon cycle equilibrium [7].

The magnitude of soil organic carbon (SOC) reservoirs is intrinsically linked to microbial activities [8], with wetland soil carbon-sequestering microbial communities specifically mediating carbon fixation and SOM formation. These microbial assemblages exhibit functional traits modulated by environmental variables, as evidenced by comprehensive metagenomic and metatranscriptomic studies [9]. For instance, CO_2_ fixation pathways (e.g., Calvin cycle, reductive TCA cycle) in wetland microbiomes demonstrate significant regulation through salinity gradients, oxygen availability, and hydrological fluctuation regimes. Soil TN content primarily drives microbial diversity, while TC determines community structure, with elevated temperature and moisture enhancing carbon fixation through specific microbial guilds [10]. Microbial communities play dual roles in organic carbon mineralization and stabilization, with their metabolic balance dictating net CO_2_/CH_4_ fluxes to the atmosphere [5]. Soil microbial community structure and function play a key role in organic carbon mineralization. Studies have shown that the composition and abundance of microbial communities and their metabolic activities directly influence the decomposition and mineralization processes of organic carbon [11]. Increased temperature accelerates the decomposition of organic matter through multiple mechanisms, including enhanced microbial activity, elevated enzyme activity, and changes in microbial community structure. In a study on mangrove soil organic matter, it was noted that global warming significantly and markedly accelerated the rate of SOM decomposition of mangrove soil organic matter [12]. Kirwan et al. found in their study that elevated CO_2_ concentration and warming may enhance the productivity of organic matter in coastal wetlands, but experiments showed that the rate of decomposition of organic matter would increase by approximately 20% for every 1 °C increase in temperature, and that this temperature-driven acceleration in the decomposition of organic matter would have an impact on sea level rise rates. This temperature-driven acceleration of organic matter decomposition will exceed the carbon sequestration effect of productivity enhancement, which may trigger a positive feedback loop of “climate warming–sea level rise–carbon release”, threatening the long-term stability of coastal wetland ecosystems and the global blue carbon function [13]. Notwithstanding, thermoadapted microbial populations may mitigate SOC depletion through metabolic acclimation mechanisms [6,14]. CO_2_-fixing microorganisms play an important role in atmospheric carbon sequestration by being able to convert CO_2_ into organic matter [15]. These microorganisms take up CO_2_ and thereby influence the renewal and cycling of organic carbon mainly through the Calvin–Benson–Bassham (CBB) cycle, which is the most common autotrophic carbon sequestration pathway and relies on the Rubisco enzyme to catalyze the formation of 3-phosphoglyceric acid (3-PGA) from the combination of CO_2_ with ribulose-1,5-bisphosphate (RuBP). Although the CBB cycle fixes about 1011 tons of CO_2_ per year, its efficiency is limited by the low carboxylation activity and competitive oxidation reactions of Rubisco [16]. In contrast, the cbbL gene, as one of the molecular biomarkers, has been widely used to study the structure and function of bacterial communities undergoing CO_2_ fixation [17].

The qinghai–Tibet Plateau, Earth’s highest and largest plateau, sequesters substantial soil carbon through cryogenic stabilization mechanisms under cold conditions [18]. Recent assessments on the QTP indicate that SOC stocks decreased by 1894 Mt C between 2000 and 2020, necessitating an enhanced carbon sink capacity through ecological restoration (e.g., wetland rehabilitation) to meet carbon neutrality targets under high-emission warming scenarios [19]. It has also been pointed out that the QTP is a critical zone for climate change [20], and climate warming affects the wetland carbon cycle of the QTP region by altering temperature [21], precipitation [21], and microbial activities [22], e.g., permafrost thawing leads to the loss of soil moisture and accelerates the mineralization of organic carbon to CO_2_ [23]. Although studies have focused on soil carbon storage in wetlands on the QTP, the dynamic response of carbon-sequestering microorganisms to warming in swampy wetlands on the QTP has not been well studied. Therefore, this study aims to assess the response of soil carbon-sequestering microorganisms to the warming of the river source wetland of the Wayan Mountains in the northeastern part of the QTP through a field simulation of warming experiments. Specifically, this study will explore the following two questions: (1) How does warming affect the diversity of cbbL carbon-fixing microorganisms? (2) How does warming affect the species composition of cbbL carbon-sequestering microbial communities?

## 2. Materials and Methods

### 2.1. Overview of the Study Area

The study area is located in the Wayan Mountains Experimental Station on the northeast shore of Qinghai Lake, which is a typical alpine wetland. The geographic location of the area is 37°43′–37°46′ N, 100°01′–100°05′ E, with an average elevation of 3720–3850 m (Figure 1). The temperature difference between day and night in this area is extremely large, with a multi-year average temperature of −3.31 °C, and the highest value occurs in July, which is 11.87 °C, and the lowest value occurs in January, which is −19.73 °C, belonging to the continental climate of the plateau; the annual precipitation is 420.37 mm. the soil type is mainly meadow soil. Affected by the altitude and climate, the vegetation type of the Wayan Mountains is relatively single, and the main dominant vegetation is Kobresia humilis [10].

### 2.2. Soil Sample Collection

The simulated warming experiment was set up in 2015 according to the International Tundra Experiment (ITEX). An open-top chamber (OTC) was set up in a 4 m × 4 m plot of 50 m × 50 m (Figure 2). The OTC consisted of six transparent glass panels made of polyacrylate (150 cm in diameter on the upper surface) [24]. Five replicates were set up, with five samples from the river source wetland of the Wayan Mountains’ natural condition (Wck) and five samples from the open-top box (WW), and the control samples were set up about 1 m away from the warming circle with a uniform texture. The device consisted of six transparent polyacrylate glass panels (upper surface diameter of 150 cm, lower surface diameter of 208 cm, top edge of 87 cm, bottom edge of 122 cm) with an inclination angle of about 60° and a light transmission of more than 92%. Soil samples were collected in early June 2020, at the beginning of the growing season. A soil auger (4.5 cm in diameter) was used to take random samples from the 0–10 cm soil layer in each sample plot by the five-point sampling method. After mixing the soil in the same layer, the sieved soil was passed through a 2 mm sieve to remove large plant roots and stones, then temporarily stored in ice packs for immediate transport to the laboratory. One portion of the soil samples was stored at −4 °C for subsequent physicochemical analysis, while the other portion was stored at −80 °C for molecular analysis. The total number of samples was 10, and the temperature increase in the thermosphere was about 1.2 °C.

### 2.3. Determination of Soil Physical and Chemical Properties

To monitor soil water content and temperature, TDR-300 (produced by Spectrum Technologies in Plainfield, IL, USA) [25] and LI8100 (manufactured by LI-COR in Lincoln, NE, USA) were used [26]. The pH probe (model FE20-FiveEasy pH, from Mettler Toledo in Gießen, Germany) was used to determine soil pH at a soil–water ratio of 1:2.5 [27]. For the determination of total carbon (TC) and total nitrogen (TN) content, an elemental analyzer (Vario EL III, Elemental Analysis System GmbH, Langenselbold, Germany) was used [28,29].

### 2.4. DNA Extraction and Illumina MiSeq Sequencing

Soil DNA was extracted using the PowerSoil DNA isolation kit (Mobio, Carlsbad, CA, USA), and the quality of soil DNA was examined by 1% concentration agarose gel electrophoresis. The cbbL gene fragments of F (5′-GACTTCACCAAAGACGACGA-3′) and R (5′-TCGAACTTGATTTCTTTCCA-3′) were amplified by using standard amplification primers for carbon-sequestering microorganisms [30]. The PCR reaction system and cycling conditions were performed as described in the literature [31]. The target strip binder was recovered and purified by Agarose Gel DNA Recovery Kit for Illumina MiSeq high-throughput sequencing. The raw data were first excised from the sequence by cut-adapt software to discard the unmatched primer sequences, and Vsearch was used to splice the sequences for quality control. Then the high-quality sequences were clustered into operational taxonomic unit (OTU) tables with 97% similarity. Finally, the sequences were annotated using the nucleotide sequence database (NT) (https://ftp.ncbi.nlm.nih.gov/blast/db/, accessed on 18 May 2025) for sequence comparison. This approach ensured the accuracy and reproducibility of the sequencing results.

### 2.5. Statistical Analysis

The grgrarecurve function in the MicrobiotaProcess package of R software (version 4.1.2) plotted dilution curves, and the get_alphain- dex function calculated the ACE index, Chao1 index, Simpson index, and Shannon index to quantify the α-diversity of the microbial communities. In order to assess the β-diversity among different samples, principal coordinate analysis (PCoA) was performed using the get_pcoa function. The aov function of the Stats package was used to perform an analysis of variance (ANOVA) to further explore the effects of different factors on the microbial community. In order to explore the correlation between the physicochemical factors, between the physicochemical factors and the microbial community, and to draw the heat map of the correlation network, the mantel_test function of the linkET package was used to calculate the correlation and *p*-value, and the heat map of the correlation network was drawn using the qcorrplot function. The upset function of the UpsetR package was used to draw the histogram of the OTUs. The Wilcoxon rank-sum test was used to test for between-group significance, and correlation heatmaps were plotted using the pheatmap package to present patterns of associations between each environmental factor and each genus level of the dominant flora. All graphs and visualizations were performed by the ggplot2 package to ensure graphical esthetics and accuracy.

## 3. Results

### 3.1. Community Diversity of cbbL Carbon-Sequestering Microorganisms in Response to Warming

With increasing sequencing depth, the WW group consistently exhibits higher values for both ACE and Chao1 indices compared to the Wck group, suggesting greater richness in the WW group (Figure 3a). The violin plots for ACE, Chao1, and Observe indices in Figure 3b reveal significant differences between the groups, with the WW group showing higher values, indicating greater species richness. For the Pielou, Shannon, and Simpson indices, no significant differences are observed between the Wck and WW groups, implying comparable evenness and diversity patterns in these aspects (Figure 3b). OTU richness analyses revealed 193 common operational taxonomic units between groups, with a total of 567 OTUs for Wck compared to 481 for WW (Figure 3c). This pattern suggests that while the Wck group was richer in microbial diversity, the WW group had a more specialized community structure. A clear separation of groups was delineated along the PC1 (56.23% of variance) and PC2 (16.19% of variance) axes in the PCA analysis (Figure 3d). The four key OTUs showed significant distributional differences along the PC1 axis, suggesting that they play a key role in driving microbial community differentiation among communities.

### 3.2. Species Composition and Functional Taxa of Carbon-Sequestering Microorganisms of cbbL in Response to Warming

The results of in-depth analysis of the cbbL carbon-sequestering microbial community in the river source wetland of the Wayan Mountains showed that the warming treatment significantly changed the structure of the community, compared with Wck, the relative abundance of Cyanobacteria in the WW group increased from 5% to 10%, and Actinobacteria went from absent to present and stabilized at 5%, whereas Proteobacteria, although still dominant, had its dominance weakened in some samples (e.g., Ww3–Ww5). In the Ww3–Ww5, its dominance was weakened, with a lower proportion in some samples than in the control group. It is worth noting that Chlorophyta maintained a stable high abundance (80%) in the WW group. In addition, a small number of “other fungi” (5%) were added to the WW group. Overall, warming may have promoted the proliferation of blue-green algae and actinomycetes, while suppressing the single dominance of Ascomycetes, with Proteobacteria being the absolutely dominant phylum in all samples, with relative abundances ranging from 53.79% to 93.68% (Figure 4a). Sixteen genus-level bacteria with relative abundance greater than 1% in the river source wetland of the Wayan Mountains were selected to construct a histogram of relative abundance percentages (Figure 4b). The simulated warming resulted in significant intergroup differences among the six genus-level microbial groups: Arthrospira, Bradyrhizobium, Ferrithrix, and Nitrosomonas had increased relative abundance, and on the contrary, Hydrogenophaga, and Rhodospirillum had decreased relative abundance (Figure 5). The highest average relative abundance was found in the Rhodospirillum of Proteobacteria and the Ferrithrix of Actinobacteria, which were 7.12% and 4.79%, respectively.

FAPROTAX functional prediction results showed that the cbbL carbon-sequestering microorganisms in the river source wetland of the Wayan Mountains were in eight major functional groups, including Phototrophy, Chemoheterotrophy, Aerobic chemoheterot-rophy, Photoautotrophy, Photosynthetic_cyanobacteria, Oxygenic_photoautotrophy, Photoheterotrophy, and the Dark oxidation of sulfur compounds (Figure 6). Among these eight functional groups, only three functional groups, chemoheterotrophy, photoheterotrophy and phototrophy, showed significant differences after simulated warming (Figure 7).

### 3.3. Changes in Environmental Factors and Their Interrelationships with cbbL Carbon-Sequestering Microbial Communities

NN, TC, and TN significantly increased in the river source wetland of the Wayan Mountains under warming treatment (Figure 8a), and the correlation network diagram further showed that phyla and genera were significantly positively correlated with NN, TN, and TC, suggesting that soil TN, TC, and NN had a significant effect on the structure of the microbial community (*p* < 0.05), and that there was a significant positive correlation between TN and TC, and a significant negative correlation between AN, and all other factors were not significantly correlated (Figure 8b). This demonstrated that the microbial community structure in the river source wetland of the Wayan Mountains was mainly regulated by NN, TN, and TC in a synergistic manner, and the independence of AN suggested that its ecological function might be regulated by other unmeasured variables. The RDA redundancy analysis further confirmed that the microbial carbon sequestration community of the cbbL was mainly influenced by NN, TN, and TC (Figure 8c). Redundancy analysis of the top 10 carbon-sequestering microbial communities and soil environmental factors showed that different environmental factors had different effects on different microorganisms, with TC having the most significant effect on microbial community structure, followed by TN, NN, and relatively weakly regulated by pH and AN. TC content was significantly and positively correlated with Thiohalomonas and Ferrithrix, which may be the most important microorganisms in the Wayan Mountains. These microorganisms may be important markers of carbon sequestration in the river source wetland of the Wayan Mountains (Figure 8d). In summary, most of the changes in microbial community structure in the river source wetland of the Wayan Mountains were significantly correlated with total carbon, total nitrogen, and nitrate nitrogen contents, with total nitrogen being the most dominant driver; however, the response of microbial community structure to pH and AN was less pronounced. Details of soil physical and chemical factors are shown in Appendix A.

## 4. Discussion

### 4.1. Effects of Simulated Warming on the Diversity of Carbon-Sequestering Microorganisms and Environmental Factors in cbbL

Bacterial community structure is predominantly regulated by temperature fluctuations, with elevated temperatures enhancing microbial biomass under stable abiotic conditions [32]. Experimental soil warming typically amplifies fungal community size while accelerating nitrogen mineralization and organic nitrogen accumulation; however, such treatments do not alter local species richness (Alpha diversity) but reduce effective species counts through dominance hierarchy restructuring [33,34]. This study revealed significant Alpha diversity enhancement (ACE/Chao1 indices) without concomitant Shannon/Simpson index alterations, consistent with Kevin et al.’s findings demonstrating fungal Alpha diversity increases under warming without evenness metric impacts [35].

Comparative analysis with Han et al.’s research showed altitude-dependent diversity responses: low-altitude warming significantly increased Shannon indices, whereas high-altitude ecosystems maintained ecological niche conservatism for evenness metrics [36]. The absence of Shannon index elevation in this study may reflect alpine-specific ecological constraints preserving evenness under warming. Soil physicochemical analyses demonstrated warming-driven NN, TC, and TN increases alongside soil moisture reductions. These carbon cycle perturbations align with Chen et al.’s observations of enhanced soil CO_2_ emissions from QTP whole-soil warming [37], while contrasting with Anne et al.’s finding that prolonged warming reduces grassland organic nitrogen mineralization rates and N_2_O effluxes through microbial activity suppression [38]. Correlation analyses identified significant phylum- and genus-level microbial community correlations with NN, TN, and TC, corroborating Dai et al.’s results [39]. However, the weak pH–microbial structure association diverges from Liao et al.’s findings [40], potentially attributable to ecosystem-specific differences in cbbL microbial community responses [41].

### 4.2. Simulated Warming Alters the Species Composition of cbbL-Containing Carbon-Fixing Microorganisms

Key flora are highly connected and they play an important role in microbial communities [42]. In this study, Proteobacteria maintained dominance across warming treatments, though their relative abundance decreased post-treatment, falling below control levels in certain experimental replicates. This phylum’s ecological resilience may stem from superior habitat adaptability, rapid proliferation rates, and efficient substrate utilization capabilities [43]. Warming induced significant Cyanobacteria proliferation while stabilizing Actinobacteria at 5% abundance. Colina et al. demonstrated that temperature-mediated Cyanobacteria biomass shifts can alter aquatic nutrient cycling and energy fluxes [44], with some thermophilic Cyanobacteria exhibiting enhanced growth competitiveness under elevated temperatures [45]. Dong et al. reported that short-term warming enhances Actinobacteria–Cyanobacteria synergies to accelerate active layer soil organic carbon decomposition, mirroring our findings [46]. However, projected Cyanobacteria expansion under climate warming raises concerns about algal bloom-induced light limitation for submerged macrophytes and phytoplankton [47]. Actinobacteria decompose complex organic matter by secreting cellulases and lignin peroxidases, which convert macromolecular organic matter into soluble sugars, providing substrates for subsequent mineralization [48]. At the genus level, Thioflexothrix, Rhodospirillum, and Ferrithrix dominated cbbL microbial assemblages in this study, contrasting with Li et al.’s findings where phagocytophilic Variovorax and root-associated Bradyrhizobium prevailed [31]. Zhang et al. investigated the microbial communities in the carbon cycle under different grazing restrictions, and the results showed that at the genus level, the microbial communities with cbbL functional genes were mainly composed of Pseudonocardia, Bradyrhizobium, and Mesorhizobium [49]. Because soil microbial community structure is influenced by a variety of environmental factors [50], changes in the ecological niche of the microbial community may also lead to changes in the dominant genera [51] so we hypothesized that this may be a reason why the dominant genera is different from the present study. Notably, Zhang et al. [28] reported Rhodospirillum dominance in Qinghai Lake wetlands, aligning with our observations. The river source wetland’s cbbL microbial composition showed environmental parameter insensitivity, potentially attributable to interactive factor buffering. Warming significantly altered six genus-level bacterial groups, among which the relative abundance of Rhodospirillum and Ferrithrix was higher, and these groups can usually be used as indicator species of the environment.

This study has several limitations. First of all, the short-term experimental duration and limited spatial replication may incompletely capture the temporal stability and spatial heterogeneity that are inherent in microbial community dynamics. For instance, previous studies documented that microbial diversity indices can exhibit significant temporal fluctuations under short-term experimental conditions, potentially reflecting transient community reorganization rather than stable ecological states [52]. Secondly, the sampling intensity might introduce biases in microbial diversity estimation, particularly for rare taxa with low spatial occupancy [53]. Thirdly, while high-throughput sequencing provides comprehensive community profiles, current methodologies remain constrained by spatial scaling effects—microbial distribution patterns often exhibit scale-dependent variability that may not be fully resolved through single-scale sampling [54]. Future research should implement longitudinal monitoring frameworks with multiple temporal checkpoints (e.g., seasonal sampling) combined with spatially explicit sampling designs spanning hydrological gradients. Such integrated approaches would enhance the resolution of both spatial heterogeneity and temporal trajectories in microbial community assembly processes.

## 5. Conclusions

This study investigated the temperature sensitivity of cbbL-harboring carbon-sequestering microbial communities through an open-top chamber simulated warming experiment in the Wayan Mountains’ river source wetland on the northeastern Qinghai–Tibetan Plateau. The results demonstrated that experimental warming significantly increased the Alpha diversity (ACE and Chao1 indices) of the microbial assemblages while altering community composition. Taxonomic analysis revealed elevated relative abundances of Cyanobacteria and Actinobacteria, concurrent with the reduced dominance of Proteobacteria. At the genus level, *Rhodospirillum* abundance decreased significantly under warming, whereas *Ferrithrix* exhibited proliferation. Functional predictions indicated enhanced heterotrophic and photoheterotrophic guilds under an elevated temperature. Soil physicochemical analyses showed warming-driven increases in NN, TC, and TN, whereas AN and pH remained statistically unchanged. Redundancy analysis identified TC as the primary environmental determinant of microbial community restructuring, followed by TN and NN. Microbial diversity metrics exhibited significant positive correlations with carbon and nitrogen indicators, confirming resource availability as the key driver of community succession. These findings enhance the understanding of microbial-mediated carbon cycling in alpine wetlands under climate warming, providing critical evidence for developing ecological protection strategies in high-altitude ecosystems.

## Figures and Tables

**Figure 1 biology-14-00708-f001:**
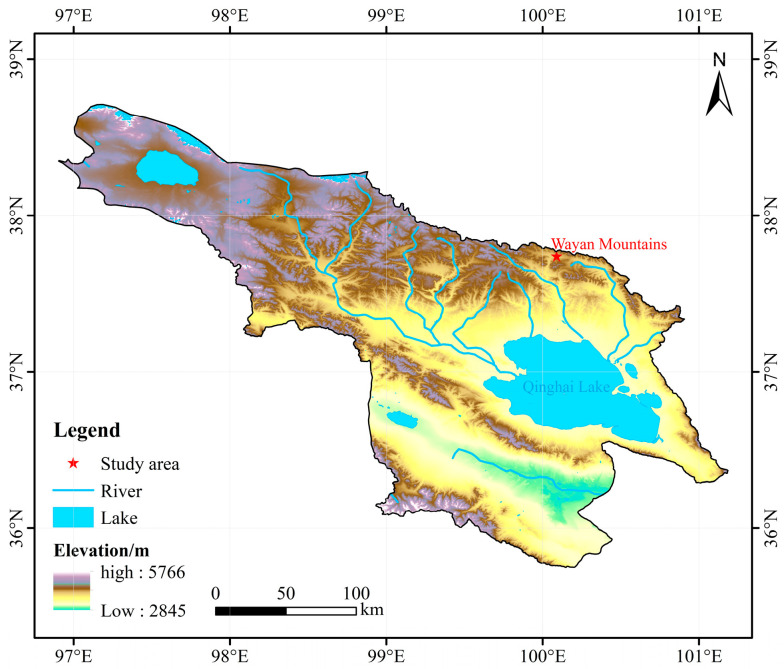
Study area overview map.

**Figure 2 biology-14-00708-f002:**
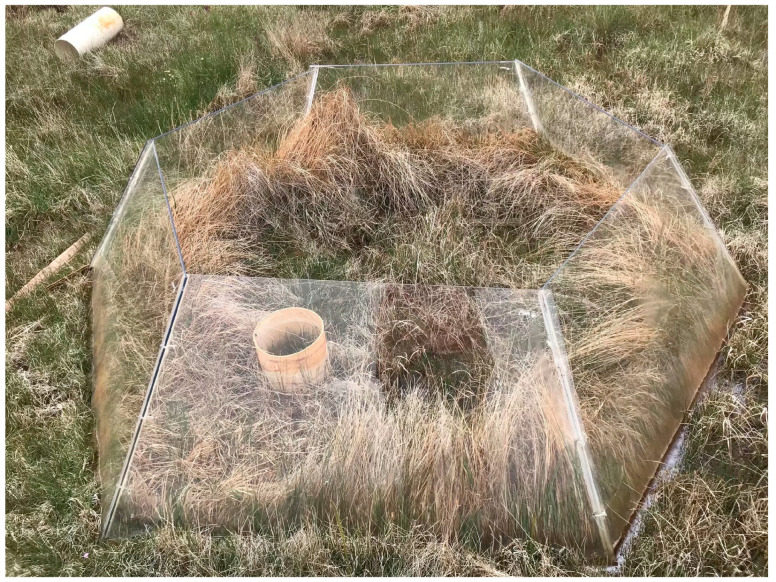
Map of treatment plots and OTC samples.

**Figure 3 biology-14-00708-f003:**
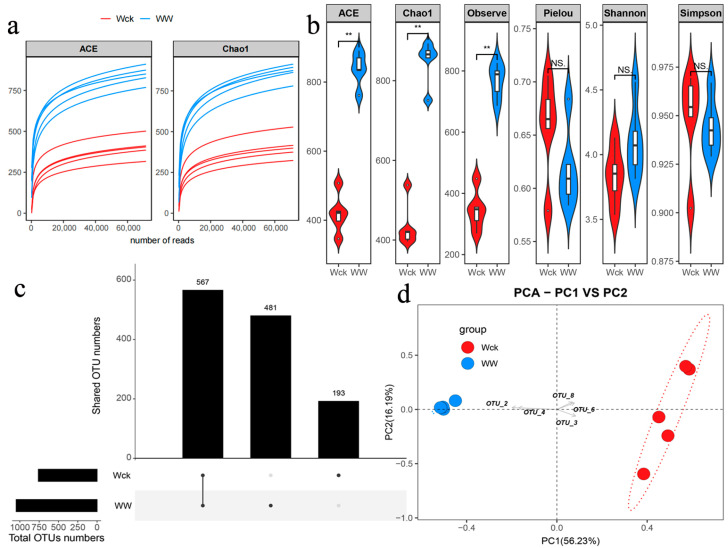
Illumina sequencing results and carbon sequestration microbial community diversity: (**a**) sample dilution curve; (**b**) cbbL microbial alpha diversity index; (**c**) OTU distribution map; (**d**) cbbL microbial principal component analysis. NS indicates *p* > 0.05, and ** indicates *p* < 0.01. Wck: samples from the natural condition; WW: samples from the open-top box.

**Figure 4 biology-14-00708-f004:**
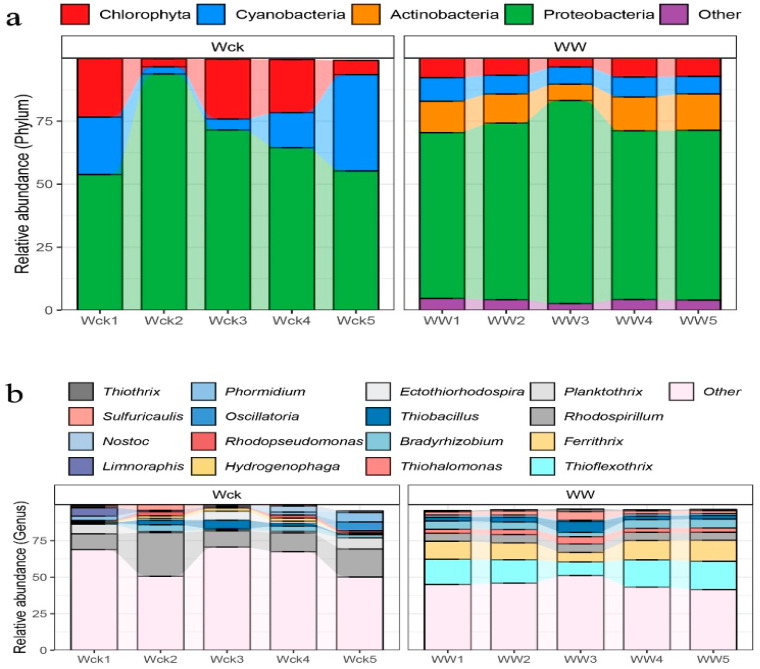
(**a**) Relative abundance and composition of cbbL carbon sequestration flora at phylum level under different treatments; (**b**) composition of genus-level communities of cbbL carbon-fixing microorganisms of Wayan Mountains. Wck: samples from the natural condition; WW: samples from the open-top box.

**Figure 5 biology-14-00708-f005:**
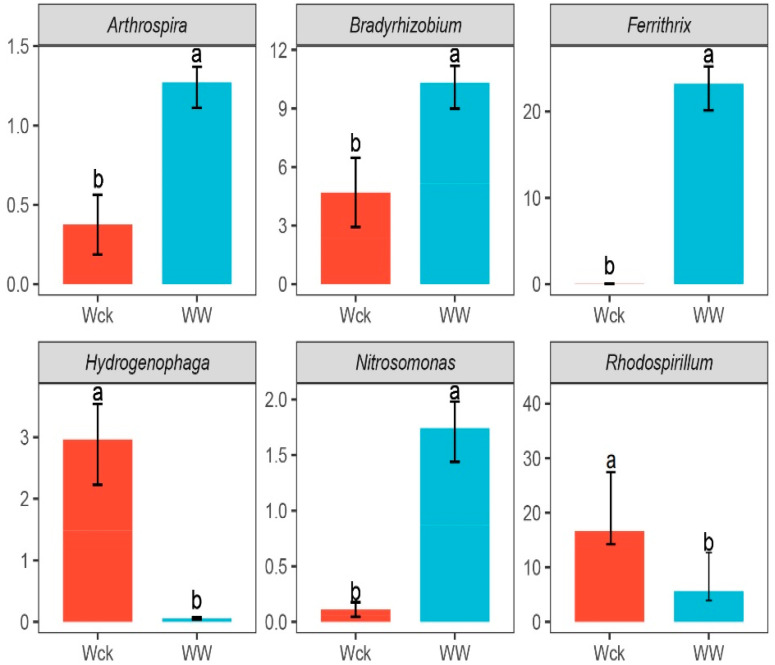
Genus-level differences in the microbiota of wetland types in the river source wetland of the Wayan Mountains. Letters a, b indicate statistical significance. Different letters indicate significant differences (*p* < 0.05). Wck: samples from the natural condition; WW: samples from the open-top box.

**Figure 6 biology-14-00708-f006:**
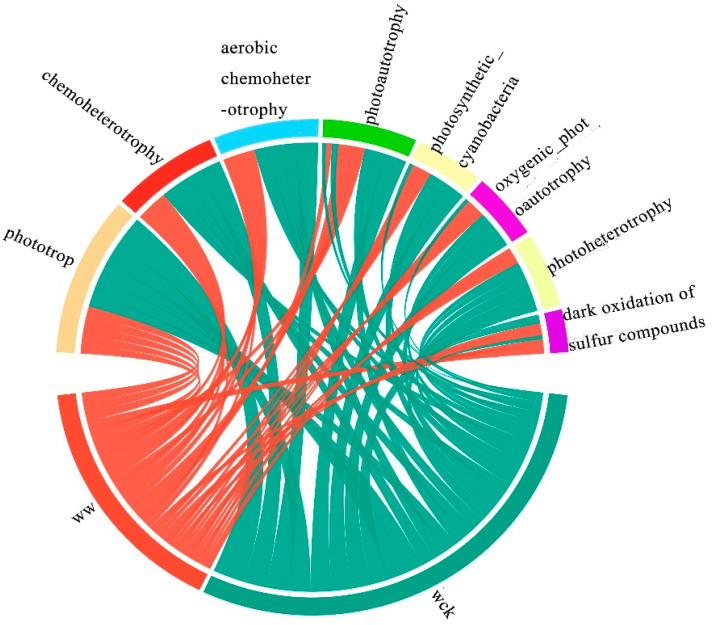
Relative abundance and composition of major functional groups of cbbL carbon-fixing bacteria under warming treatment. Wck: samples from the natural condition; WW: samples from the open-top box.

**Figure 7 biology-14-00708-f007:**
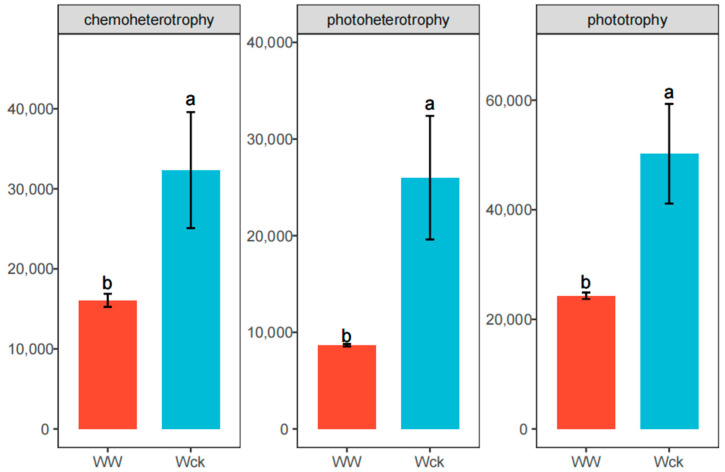
Functional taxa with significant differences in cbbL carbon sequestration bacteria under warming treatments. Letters a, b indicate statistical significance. Different letters indicate significant differences (*p* < 0.05). Wck: samples from the natural condition; WW: samples from the open-top box.

**Figure 8 biology-14-00708-f008:**
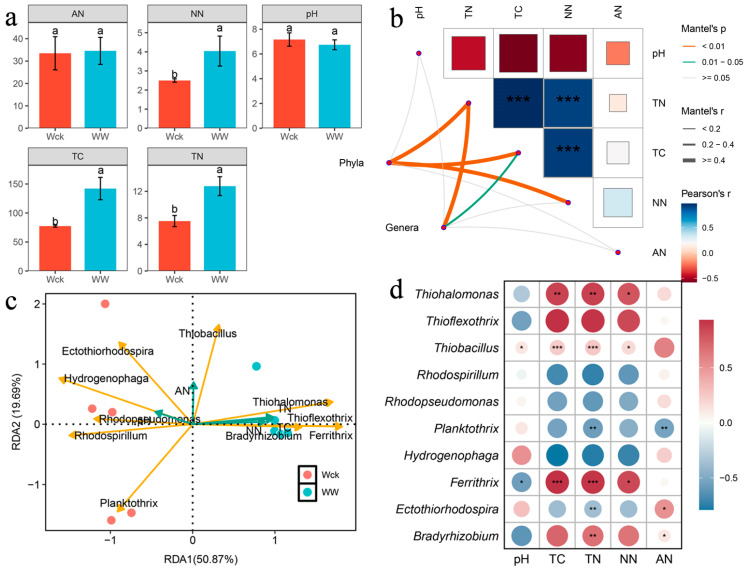
Correlation between environmental factors and carbon-sequestering microorganisms in wetland soils of the Wayan Mountains: (**a**) variation in physicochemical factors in different types of wetlands; (**b**) network diagram of correlation between carbon-sequestering microorganisms’ community characteristics and environmental factors; (**c**) redundancy analysis of environmental factors and genera of microorganisms’ communities (Top 10); and (**d**) heatmap of correlation of environmental factors and genera of microorganisms’ communities (Top 10). Letters a, b denote significance, the same letter denotes non-significant differences between groups (*p >* 0.05), and different letters denote significant differences between groups (*p* < 0.05); * denotes *p* < 0.05, ** denotes *p* < 0.01, *** denotes *p* < 0.001.

## Data Availability

The raw data have been uploaded to NCBI, and its BioProject is PRJNA1210844.

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
