# Peer review of "Response of cbbL Carbon-Sequestering Microorganisms to Simulated Warming in the River Source Wetland of the Wayan Mountains"

_biology, 2025, doi:10.3390/biology14060708_

Round 1

Reviewer 1 Report

Comments and Suggestions for Authors

The reviewed manuscript, ‘Response of cbbL carbon sequestering microorganisms to simulated warming in the river source wetland of the Wayan Mountains’ by S. Zhou et al., presents the results of an assessment of the responses of microorganisms that bind carbon in the soil of wetlands in the Wayan Mountains to experimental conditions simulating warming.

Despite the limitations of this work mentioned by the authors themselves in lines 366–380, this study can undoubtedly contribute to our understanding of the impact of modern climate change on ecosystems of various scales. The manuscript may be of interest to a wide range of readers.

The manuscript is well structured; a large amount of data has been collected and processed. The data has been processed using modern statistical methods. The results of the work are excellently visualised. The literature cited in the manuscript is relevant. We recommend that the authors supplement the Introduction section with additional modern sources on the research topic, which will help to expand the list of references.

Сomments:

Introduction: Please provide a reference to support this statement (lines 116–118).

For Section 2.1, it would be useful to include a map of the study area.

Section 2.2: Could you please add a reference for lines 141–142?

It would also be useful to include a photograph of the open-top chamber (OTC) in this section.

For lines 154-156, please specify how long the samples were stored before processing.

For lines 166–168, please specify which standards were used and add this information to the list of references.

In Figure 3, you state that 'the same letter indicates non-significant differences between groups (p > 0.05)', yet there are no identical letters in the figure. This caption is probably unnecessary. It also raises the question of why the letter 'a' corresponds to both Wck and WW. The same applies to the letter 'b'. In my opinion, since figures can be considered separate objects, the full names of the abbreviations used – Wck and WW, etc. – should be indicated in the captions. This is despite them being explained in the text.

The manuscript can be published once the author has made the necessary corrections.

Author Response

Thanks to the reviewer for their meticulous revisions and suggestions, which greatly improved the scientific research level of this paper. We all agree with these comments and have modified it with reference to comments.

Revisions in the manuscript are highlighted in red.

1) Introduction: Please provide a reference to support this statement (lines 116–118).

We have added relevant references to what is stated in lines 116-118 and appreciate your valuable comments!

2) For Section 2.1, it would be useful to include a map of the study area.

We have added a map of the study area in 2.1.

3) Section 2.2: Could you please add a reference for lines 141–142?

We have added references in lines 141-142, which are highlighted in red in the manuscript.

4)It would also be useful to include a photograph of the open-top chamber (OTC) in this section.

We have added photos of the OTC in 2.2.

5) For lines 154-156, please specify how long the samples were stored before processing.

Since this is too redundant in the text, we explain it here: soil samples for physicochemical analyses are air-dried in the lab for three days before experiments are performed, and soil samples for molecular analyses are stored for one day in a refrigerator at -80 degrees Celsius before being wrapped in dry ice and sent out for analysis by a biological company.

6) For lines 166–168, please specify which standards were used and add this information to the list of references.

We have modified this as requested.

7)In Figure 3, you state that 'the same letter indicates non-significant differences between groups (p > 0.05)', yet there are no identical letters in the figure. This caption is probably unnecessary. It also raises the question of why the letter 'a' corresponds to both Wck and WW. The same applies to the letter 'b'. In my opinion, since figures can be considered separate objects, the full names of the abbreviations used – Wck and WW, etc. – should be indicated in the captions. This is despite them being explained in the text.

1.We have removed the same letters in the figure names to show that the differences between groups are not significant (p > 0.05).

2.The letters a and b indicate only between-group variability, and different letters indicate significant differences (p < 0.05), e.g.,  in the Arthrospira group, Wck (red columns) is labeled “b” and WW (blue columns) is labeled “b”. In the Arthrospira group, Wck (red column) is labeled “b” and WW (blue column) is labeled “a”, indicating a significant difference in abundance; in the Hydrogenophaga group, Wck is labeled “a In the Hydrogenophaga group, Wck is labeled as “a” and WW is labeled as “b”, also indicating significant differences between groups. These letters are not intrinsic properties of the corresponding treatment groups, but are used to visualize the results of comparisons between different groups. The fact that the same letter appears on the bars of different treatment groups indicates that they belong to the same statistical significance category.

3.We have added the full name of the abbreviation to each figure title.

Reviewer 2 Report

Comments and Suggestions for Authors

Review of the manuscript of the article Shijia Zhou, Kelong Chen, Ni Zhang, Siyu Wang, Zhiyun Zhou and Jianqing Sun «Response of cbbL carbon sequestering microorganisms to simulated warming in the river source wetland of the Wayan Mountains».

General comments. The topic of the reviewed work is one of the most popular and relevant. It is devoted to the analysis of the influence of temperature and other abiotic factors on the taxonomic and functional diversity of soil cbbL carbon sequestering microorganism communities. It was carried out using modern high-tech methods on very interesting ecological objects - high-mountain wetland ecosystems of the Qinghai-Tibet Plateau. It is also worth noting that the data were obtained in simulated warming experiments, which makes this work especially interesting. Therefore, it undoubtedly deserves to be published.

Detail comments. However, when analyzing the manuscript, some questions arise.1. Section 2.2.Soil Sample Collection briefly but clearly describes the design of the chambers, as well as the methodology for taking soil samples, but there is no information on how many chambers there were or on what principle they were installed.There is also no clarity regarding the duration of the experiment: it is said that it began in 2011, and soil samples were taken at the beginning of the 2020 growing season: does this mean that the experiment lasted 9 years?It is unclear when the control soil samples characterizing the original microorganism communities were taken, how often the temperature, humidity, pH, and carbon and nitrogen content of the soil were measured?I consider it necessary to clearly state all these questions!2. There are complaints about the presentation of the obtained results.For example, the size of all the figures is frankly unsuccessful - they are very small, difficult to read!In addition, in the text of the work and in the figures there are a large number of various abbreviations, perhaps for those who constantly work with them, this does not create any problems, but for other specialists this is already a problem, complicating the understanding of the essence of the work.In particular, in the figures there are such abbreviations as WW and Wck, but their explanations could not be found!Looking at the figures, it is difficult to understand where the control characteristics are, and where the experimental ones.The captions to figures 2 and 3 are detailed, and to figures 4 and 5 are brief, although in both cases they should be of the same type: in some, the dynamics of taxonomic diversity is shown, and in others, functional diversity.

I believe that after correcting the comments made, the manuscript can be accepted for publication.

Review of the manuscript of the article Shijia Zhou, Kelong Chen, Ni Zhang, Siyu Wang, Zhiyun Zhou and Jianqing Sun «Response of cbbL carbon sequestering microorganisms to simulated warming in the river source wetland of the Wayan Mountains».

General comments. The topic of the reviewed work is one of the most popular and relevant. It is devoted to the analysis of the influence of temperature and other abiotic factors on the taxonomic and functional diversity of soil cbbL carbon sequestering microorganism communities. It was carried out using modern high-tech methods on very interesting ecological objects - high-mountain wetland ecosystems of the Qinghai-Tibet Plateau. It is also worth noting that the data were obtained in simulated warming experiments, which makes this work especially interesting. Therefore, it undoubtedly deserves to be published.

Detail comments. However, when analyzing the manuscript, some questions arise.1. Section 2.2.Soil Sample Collection briefly but clearly describes the design of the chambers, as well as the methodology for taking soil samples, but there is no information on how many chambers there were or on what principle they were installed.There is also no clarity regarding the duration of the experiment: it is said that it began in 2011, and soil samples were taken at the beginning of the 2020 growing season: does this mean that the experiment lasted 9 years?It is unclear when the control soil samples characterizing the original microorganism communities were taken, how often the temperature, humidity, pH, and carbon and nitrogen content of the soil were measured?I consider it necessary to clearly state all these questions!2. There are complaints about the presentation of the obtained results.For example, the size of all the figures is frankly unsuccessful - they are very small, difficult to read!In addition, in the text of the work and in the figures there are a large number of various abbreviations, perhaps for those who constantly work with them, this does not create any problems, but for other specialists this is already a problem, complicating the understanding of the essence of the work.In particular, in the figures there are such abbreviations as WW and Wck, but their explanations could not be found!Looking at the figures, it is difficult to understand where the control characteristics are, and where the experimental ones.The captions to figures 2 and 3 are detailed, and to figures 4 and 5 are brief, although in both cases they should be of the same type: in some, the dynamics of taxonomic diversity is shown, and in others, functional diversity.

I believe that after correcting the comments made, the manuscript can be accepted for publication.

Author Response

Thanks to the reviewer for their meticulous revisions and suggestions, which greatly improved the scientific research level of this paper. We all agree with these comments and have modified it with reference to comments.

Revisions in the manuscript are highlighted in red.

1)Soil Sample Collection briefly but clearly describes the design of the chambers, as well as the methodology for taking soil samples, but there is no information on how many chambers there were or on what principle they were installed.

Number of Chambers: We clarified that a total of 10 chambers were deployed in the study (5 treatment chambers and 5 control chambers). This information is now explicitly stated in Section 2.2.

2)There is also no clarity regarding the duration of the experiment: it is said that it began in 2011, and soil samples were taken at the beginning of the 2020 growing season: does this mean that the experiment lasted 9 years?

2011 in the manuscript was indeed an error; we actually established the OTC in 2015, the experiment lasted 5 years, and a sampling was conducted in June 2020 to determine microbial and soil physicochemical properties.

3)The size of all the figures is frankly unsuccessful - they are very small, difficult to read!

We have rearranged the image size so that the text on the image is clearly visible.

4)In particular, in the figures there are such abbreviations as WW and Wck, but their explanations could not be found! Looking at the figures, it is difficult to understand where the control characteristics are, and where the experimental ones. The captions to figures 2 and 3 are detailed, and to figures 4 and 5 are brief, although in both cases they should be of the same type: in some, the dynamics of taxonomic diversity is shown, and in others, functional diversity.

We've added explanations of abbreviations in chart titles and explanations of charts.

Reviewer 3 Report

Comments and Suggestions for Authors
  1. Analysis limited to 0–10 cm soil. Deeper layers (affected by permafrost thaw) may influence carbon cycling. If possible, ddress this limitation.
  2. Suggest providing a comparison database.
  3. Vegetation may have an impact on data.
  4. Aseptic operation during the sampling process.
  5. If there are more localization parameters, it will enhance persuasiveness.
  6. Correct "the source the river source wetland" (p.6) to "the river source wetland."

Author Response

Thanks to the reviewer for their meticulous revisions and suggestions, which greatly improved the scientific research level of this paper. We all agree with these comments and have modified it with reference to comments.

Revisions in the manuscript are highlighted in red.

1)Analysis limited to 0–10 cm soil. Deeper layers (affected by permafrost thaw) may influence carbon cycling. If possible, ddress this limitation.

Thank you for pointing this out. We agree with this comment. Therefore, we chose to sample during the pre-growing season, thus avoiding the seasonal freezing period

2)Suggest providing a comparison database.

Thanks to your valuable comments, we used the Nucleotide Sequence Database (NT) (ftp://ftp.ncbi. nih. gov/blast/db/) for sequence comparison annotation.Explained in line 183 of section 2.4 of the manuscript

3)Vegetation may have an impact on data.

Thank you for your interest in the potential impact of vegetation on our data. On the Tibetan Plateau, sampling in early June occurs during the low-activity early growing season: temperatures are still low, vegetation cover is sparse and mostly dormant or just sprouting, and biomass is minimal. Therefore, the direct effect of vegetation on soil parameters was minimal.

4)Aseptic operation during the sampling process.

Each step in our sampling process is strictly alcohol-sterilized, and the steps in the references are strictly followed to ensure the accuracy of sampling.

5)If there are more localization parameters, it will enhance persuasiveness.

We have added a map of the study area in 2.1.

6)Correct "the source the river source wetland" (p.6) to "the river source wetland."

We have fixed a typo on page 6, line 230. in the river source wetland of the Wayan Mountains
